# Distribution Adaptation and Classification Framework Based on Multiple Kernel Learning for Motor Imagery BCI Illiteracy

**DOI:** 10.3390/s22176572

**Published:** 2022-08-31

**Authors:** Lin Tao, Tianao Cao, Qisong Wang, Dan Liu, Jinwei Sun

**Affiliations:** School of Instrumentation Science and Engineering, Harbin Institute of Technology, Harbin 150001, China

**Keywords:** BCI illiteracy, multiple kernels learning, maximum mean discrepancy, extreme learning machine, random forest

## Abstract

A brain-computer interface (BCI) translates a user’s thoughts such as motor imagery (MI) into the control of external devices. However, some people, who are defined as BCI illiteracy, cannot control BCI effectively. The main characteristics of BCI illiterate subjects are low classification rates and poor repeatability. To address the problem of MI-BCI illiteracy, we propose a distribution adaptation method based on multi-kernel learning to make the distribution of features between the source domain and target domain become even closer to each other, while the divisibility of categories is maximized. Inspired by the kernel trick, we adopted a multiple-kernel-based extreme learning machine to train the labeled source-domain data to find a new high-dimensional subspace that maximizes data divisibility, and then use multiple-kernel-based maximum mean discrepancy to conduct distribution adaptation to eliminate the difference in feature distribution between domains in the new subspace. In light of the high dimension of features of MI-BCI illiteracy, random forest, which can effectively handle high-dimensional features without additional cross-validation, was employed as a classifier. The proposed method was validated on an open dataset. The experimental results show that that the method we proposed suits MI-BCI illiteracy and can reduce the inter-domain differences, resulting in a reduction in the performance degradation of both cross-subjects and cross-sessions.

## 1. Introduction

A brain–computer interface (BCI) based on electroencephalography (EEG) enables a user to control external devices by decoding brain activities that reflect the user’s thoughts [1]. For example, a user’s motor imagery (MI) can be translated into external device control by an MI-BCI. Some subjects cannot effectively control BCI equipment, meaning that they achieve a classification accuracy of less than 70%; such subjects are referred to as BCI illiterate [2,3]. Poor repeatability is obvious with MI-BCI, which can elicit SMR underpinned by neurophysiological processes [4,5]. As shown in Figure 1, the power spectral density (PSD) of Subject 46 was quite different in each of the two sessions. However, it is generally assumed that the training samples and test samples followed the same statistical distribution when a BCI system is based on machine learning. Domain adaptation (DA), as it pertains to transfer learning, has proven to be an effective method to handle inter-domain shift [6]. To make the distribution of features between the source domain and target domain become even closer to each other, we need to adapt both the marginal distribution and the conditional distribution. Furthermore, MI-BCI illiterate subjects do not display typical brain events such as event-related desynchronization (ERD) and event-related synchronization (ERS) [7]. Their divisibility of features was low, as is shown in Figure 2. Therefore, when studying the shared model of the source domain and target domain, the maximum divisibility of features and the impact of a low classification rate should be taken into consideration alongside the inter-domain shift.

The goal of feature-based marginal distribution adaptation (MDA) methods is to have a common feature space in which the marginal distribution of the source domain and the target domain are as close as possible. Certain achievements have been made in the adaptation of marginal distribution in many fields including EEG signal processing with these methods. Liu et al. [9] applied transfer component analysis (TCA) in EEG-based cross-subject mental fatigue recognition. Zhang et al. [10] reduced distribution differences through an inter-domain scatter matrix based on cross-subject mental workload classification. Chai et al. [11] proposed the use of subspace alignment (SA) to transform features into a domain-invariant subspace to solve the adaptation problem of EEG-based emotion recognition. He et al. [12] applied correlation alignment (CORAL) to minimize the spatial offset when solving the problem of different set domain adaptation for BCIs. Hua et al. [13] applied geodesic flow kernel (GFK) in EEG-based cross-subject emotion recognition. With the development of transfer learning, new progress has also been made in MDA. Wei et al. [14] applied the linear weighting method to the four frequently adopted DA methods (TCA, manifold alignment, CORAL, and SA) to determine coefficients through repeated iterations using the principle of neighborhood consistency. Ma et al. [15] identified the center particle position between the two domains and aligned the center position of the source domain and the target domain by translation. A common problem of these approaches is that although they reduce the marginal distribution differences between the source domain and the target domain in the new subspace, the data categories remain indistinguishable, as displayed in Figure 3a. Considering the difficulties in classifying the features extracted by the MI-BCI illiterate subjects, the above works were probably not optimal for BCI illiteracy. We need to find a new subspace in which the divisibility of categories is maximized and the difference between domains is minimized, as displayed in Figure 3b.

Inspired by the kernel trick that data can be mapped to high-dimensional space to increase data divisibility, the kernel method provides a powerful prediction framework for learning nonlinear prediction models. Therefore, we attempted to map features to the Reproducing Kernel Hilbert Space (RKHS) to find latent features of the subjects, especially BCI illiterate subjects, in this multi-dimensional nonlinear space to improve the class divisibility. Meanwhile, to address the problem that a single kernel has relatively more limitations for wide feature distribution, instead, we applied the linear combination of a series of basic kernels. The combined kernel function could still satisfy the Mercer condition, that is, the function satisfies the symmetry and positive definiteness [16,17]. Multiple kernel based maximum mean discrepancy (MK-MMD) put forward in [18] maps both the source domain and target domain data to multiple-kernel-based RKHS and then minimizes the center distance to reduce the marginal distribution difference. Following this idea, we adopted multiple kernel learning (MKL) combined with MK-MMD to build a DA framework. DA combined with MKL was then addressed using the classifier-based DA method in many studies [19,20,21,22,23]. The method involved adding the objective function that minimized the distance between domains in the mapped feature space to the risk function with a kernel-based classifier and applied a weight parameter λ to balance the data distribution differences between the two domains and structural risk functions. This method demonstrated improved classification and generalization capabilities. However, Chen et al. [24] pointed out that the results of minimizing the risk function with the above methods depend on the parameter λ, which may sometimes sacrifice domain similarity to achieve a high classification accuracy for the source only. Zhang et al. [25] put forward a marginal distribution adaptive framework for kernel-based learning machines. This framework first maps the original features to RKHS to improve the divisibility of categories and then transfers the original data to the target domain through linear operators in the result space to make the processed data become close to the covariance of the target data. Inspired by this method, we proposed a distribution adaptation framework based on multiple kernel. Specifically, we used the source domain data to train the multiple-kernel extreme learning machine (MK-ELM) [26] and found that the multiple-kernel induced RKHS, which can maximize the divisibility of source domain feature categories. We then applied MK-MMD to align the source domain and the target domain under this result space. Features after transformation can retain the information of the original data as much as possible [27,28,29]. Therefore, the proposed method can achieve the maximal divisibility of categories and minimal shift between domains.

It is necessary to retain as much information as possible for MI-BCI illiteracy during feature extraction, so the dimension of features will be relatively high. In light of this point, we applied random forest (RF) as the classifier, which can be used without dimensionality reduction and cross validation. RF has been widely used in the field of BCI and has achieved good results [30,31,32].

Considering MI-BCI illiteracy and referring to the existing technology, we proposed a framework combined distribution adaptation with RF based on multiple kernel learning (MK-DA-RF). We then verified this framework using an open dataset containing BCI illiterate subjects [8]. The main contributions of this study are as follows.

The source domain data were applied to train kernel-based ELM to find a subspace that could achieve the best classification effect, that is, the separability of features was the best in this new subspace;To overcome the limitations of a single kernel, a linear connection framework using multiple basic kernels was proposed;MK-MMD was applied to align the distribution of the mapped source and target domain data in this subspace.

The rest of this paper is organized as follows. Section 2 introduces the related work and methods used in this study. The experimental results and discussion are provided in Section 3 and Section 4, respectively. Section 5 presents the conclusions of the study.

## 2. Methodology

Our proposed distribution adaptation and classification framework based on multi-kernel learning is displayed in Figure 4. The extracted features of the EEG-based BCI system used for the training classifier were defined as source domain features, and the features used for testing were defined as target domain features. The source domain features were used to train the multiple-kernel ELM and the weights were then determined. Next, the kernel function that allows for the maximal inter-class divisibility after the data are mapped to the new RKHS is obtained. Then, the features of the source domain and the target domain were aligned based on MK-MMD under the new RKHS. Then, we applied the adapted training features to train the RF to obtain a suitable classifier.

In this study, $\left\{X^{S},\;T^{S}\right\}$ is the labeled source domain data, where $X^{S}\in R^{D\times N_{S}}$ is the source domain data with *D* as the data dimension, $N_{S}$ is the number of data points in the source domain, and $T^{S}\in R^{1\times N_{S}}$ is the corresponding label; $\left\{X^{T}\right\}$ is defined as the unlabeled target domain data, where $X^{T}\in R^{D\times N_{T}}$ is the source domain data with $N_{T}$ as the number of data points in the source domain without available labels; and Class C is contained in both the source domain and target domain.

### 2.1. Distribution Alignment Based on Multiple Kernel

The goal of DA is to have equal probability densities of distributions in the new subspace between the source and target domains, in other words, $P\left(\varphi \left(X^{S}\right)\right)\approx \left(\varphi \left(X^{T}\right)\right)$.

#### 2.1.1. Multiple Kernel Expression

To overcome the limitations of a single kernel, a linear connection framework using multiple basic kernels is proposed. The mathematical expression is defined as follows:(1a)$$\varphi \left(\cdot ,\gamma \right)=\overset{m}{\underset{p=1}{\sum }}\sqrt{\gamma_{p}}\varphi_{p}\left(\cdot \right),\;\;p=1,\;2,\ldots ,k\;$$
(1b)$$K\left(\cdot ,\cdot ;\gamma \right)=\overset{m}{\underset{p=1}{\sum }}\gamma_{p}k_{p}\left(\cdot ,\cdot \right),\;p=1,2,\ldots ,k$$
where $\varphi_{p}\left(\cdot \right)$ refers to the basic mapping function; $k_{p}\left(\cdot ,\cdot \right)$ is the corresponding kernel function; and $\gamma_{p}$ is the coefficient.

#### 2.1.2. Multiple-Kernel Extreme Learning Machine

Assume that there are *N* training samples, in other words, $\left\{X,\;T\right\}=\left\{x_{i},\;t_{i}\right\},\;i=1,\;2,\;\ldots ,\;N$. According to the research by Huang et al. [33], the output of kernel based ELM for binary classification is:(2)$$y=sign\left(h\left(x\right)\beta \right)$$
where β refers to connection weight, and $h\left(x\right)$ refers to the feature mapping function.

The learning objective of ELM is to minimize the learning error and weight coefficient, which can be expressed as:$$\mathrm{Minimize}:\;L_{P_{ELM}}=\frac{1}{2}\beta^{2}+C\frac{1}{2}\overset{N}{\underset{i=1}{\sum }}{||\xi_{i}||}^{2}$$
(3)$$\mathrm{Subject}\;\mathrm{to}:h(x_{i})\beta =t_{i}-\xi_{i},i=1,2,\ldots ,N$$
where $\xi_{i}$ is the training error and C is a parameter set by the user and provides a tradeoff between the output weights and training error.

Based on the Karush–Kuhn–Tucker (KKT) theory and Bartlett’s theory [34], the Lagrangian function can be written as follows:(4)$$L_{D_{ELM}}=\frac{1}{2}{||\beta ||}^{2}+C\frac{1}{2}\overset{N}{\underset{i=1}{\sum }}{||\xi_{i}||}^{2}-\overset{N}{\underset{i=1}{\sum }}\alpha_{i}\left(h(x_{i}\right)\beta -t_{i}+\xi_{i})$$

According to the solution method of the KKT and Mercer’s theorem, we set the derivative of (4) with regard to the parameters, which can be expressed as:(5a)$$\frac{\partial L_{D_{ELM}}}{\partial \beta_{j}}\;=\;0\to \beta_{j}\;=\;\overset{N}{\underset{i=1}{\sum }}\alpha_{i,j}h{\left(x_{i}\right)}^{T}\to \beta =H^{T}\alpha$$
(5b)$$\frac{\partial L_{D_{ELM}}}{\partial \xi_{i}}=0\to \alpha_{i}=C\xi_{i},\;i=1,2,‥\ldots ,N$$
(5c)$$\frac{\partial L_{D_{ELM}}}{\partial \alpha_{i}}=0\to h(x_{i})\beta -t_{i}^{T}+\xi_{i}^{T}=0,\;i=1,2,‥\ldots ,N$$

Subsequently, by substituting Equation (5a,b) into Equation (5c), it can thus be inferred that
(6)$$\left(\frac{I}{C}+HH^{T}\right)\alpha =T$$

By substituting Equation (6) in Equation (5a),
(7)$$\beta \;=\;H^{T}{\left(\frac{I}{C}+HH^{T}\right)}^{-1}T,$$

Combining (2) with (7), the relationship between the input and output of kernel-based ELM can be expressed as:(8)$$f\left(x\right)=\;\left[\begin{array}{c} K\left(x,x_{1}\right)\\ \vdots \\ K\left(x,x_{N}\right) \end{array}\right]{\left(\frac{I}{C}+\Omega_{\mathrm{ELM}}\right)}^{-1}T$$
where:$$\Omega_{ELM}=HH^{T}:\;\Omega_{ELM_{i,j}\;}=\;K\left(x_{i},x_{j}\right)=\;h\left(x_{i}\right)\cdot h\left(x_{j}\right)$$

On this basis, the single-kernel linear combination is replaced by a multiple-kernel linear combination. The target function of MK-ELM is gained combined with (1) [26]. It can be expressed as:$$\underset{\gamma }{\min}\underset{\beta ,\xi }{\min}\frac{1}{2}{||\beta ||}_{F}^{2}+\frac{C}{2}\overset{n}{\underset{i=1}{\sum }}{||\xi_{i}||}^{2}$$
$$s.t.\beta^{T}\varphi \left(x_{i};\gamma \right)=t_{i}\;-\;\xi_{i},\;\forall i$$
(9)$$\overset{m}{\underset{p=1}{\sum }}\gamma_{p}^{q}=1,\;\gamma_{p}\;\geq \;0,\;\forall p$$

Herein, *q = 2*, $\tilde{\beta }=\left[{\tilde{\beta }}_{1},\;{\tilde{\beta }}_{2},\ldots ,\;{\tilde{\beta }}_{m}\right]$, and ${\tilde{\beta }}_{p}=\sqrt{\gamma_{p}}\beta_{p},\;p=1,\;2,\ldots \;,\;m$. The Lagrangian function is:(10)$$L\left(\tilde{\beta },\xi ,\gamma \right)=\frac{1}{2}\overset{m}{\underset{p=1}{\sum }}\frac{||{\tilde{\beta }}_{p}||{}_{F}^{2}}{\gamma_{p}}+\frac{C}{2}\overset{n}{\underset{i=1}{\sum }}{||\xi_{i}||}^{2}-\overset{T}{\underset{t=1}{\sum }}\overset{n}{\underset{i=1}{\sum }}\alpha_{it}\left(\overset{m}{\underset{p=1}{\sum }}{\tilde{\beta }}_{p}{}^{T}\varphi_{p}\left(x_{i}\right)\;-\;t_{ti}+\xi_{ti}\right)+\tau (\overset{m}{\underset{p=1}{\sum }}\gamma_{p}^{2}\;-\;1)$$

According to the KKT theory, it can be concluded that:(11)$$||{\tilde{\beta }}_{p}||{}_{F}=\gamma_{p}\sqrt{\overset{T}{\underset{s,t=1}{\sum }}\overset{n}{\underset{i,j=1}{\sum }}\alpha_{it}\alpha_{js}K_{p}\left(x_{i},x_{j}\right)}$$

By taking the derivative of (10) with respect to $\gamma_{p}$, we obtain:(12)$$-\frac{1}{2}\frac{||{\tilde{\beta }}_{p}||{}_{F}^{2}}{\gamma_{p}^{2}}+q\tau \gamma_{p}^{q\;-\;1}=0,\;p=1,\ldots m$$

Combining (11) with $\sum_{p=1}^{m}\gamma_{p}^{q}=1$, we obtain:(13)$$\gamma_{p}=\frac{||{\tilde{\beta }}_{p}||{}_{F}^{2/\left(1+q\right)}}{{\left(\sum_{p=1}^{m}||{\tilde{\beta }}_{p}||{}_{F}^{2q/\left(1+q\right)}\right)}^{1/q}},\;\forall p$$

By gradual iteration coefficient, $\gamma^{new}$ is updated and the optimal coefficient is obtained.

#### 2.1.3. Multiple Kernel Maximum Mean Discrepancy

It is assumed that $P\left(X_{S}\right)\;\neq \;P\left(X_{T}\right)$; however, there is a mapping φ so that $P\left(\varphi \left(X_{S}\right)\right)=P\left(\varphi \left(X_{T}\right)\right)$. The MMD put forward in the research by Pan et al. [27] is often used as an indicator for calculating the distribution distance in the RKHS:(14)$$Dist_{K}\left(X_{s},\;X_{T}\right)=∥\frac{1}{n_{S}}\overset{n_{S}}{\underset{i=1}{\sum }}\varphi \left(x_{Si}\right)\;-\;\frac{1}{n_{T}}\overset{n_{T}}{\underset{i=1}{\sum }}\varphi \left(x_{Ti}\right){∥}_{H}^{2}$$

In combination with (1), the multiple basic mapping functions can be regarded as one mapping function after linear combination (i.e., the mapping function is defined as $\varphi =\varphi \left(\cdot ,\gamma \right)=\sum_{p=1}^{m}\sqrt{\gamma_{p}}\varphi_{p}\left(\cdot \right)$) and a single kernel based MMD can form MK-MMD in this way.

If $X=\left\{X_{S},\;X_{T}\right\}\in R^{D\times \left(n_{S}+n_{T}\right)}$, then the kernel mapping is $\varphi \left(X\right)=\{\varphi \left(x_{1}\right),\;\varphi \left(x_{2}\right),\ldots ,$ $\varphi \left(x_{n_{S}+n_{T}}\right)\}$ and the kernel matrix is $K=\varphi {\left(X\right)}^{T}\varphi \left(X\right)$. According to the Kernel PCA theory [27,35], the transformed features can be expressed as $Z=W^{T}\varphi {\left(x\right)}^{T}\varphi \left(x\right)=W^{T}K$, where *W* is the kernel-PCA transformation matrix. By definition, this vector can retain the maximal mapped feature space information, and (14) is then written as:(15)$$Dist_{K}\left(X_{s},\;X_{T}\right)=∥\frac{1}{n_{S}}\overset{n_{S}}{\underset{i=1}{\sum }}W^{T}K_{i}\;-\;\frac{1}{n_{T}}\overset{n_{S}+n_{T}}{\underset{j=n_{S\;}+1}{\sum }}W^{T}K_{j}{∥}_{H}^{2}=tr\left(W^{T}KLKW\right)$$
where
$$L_{ij}=\left\{\begin{array}{c} \begin{array}{cc} \frac{1}{n_{s}^{2}} & x_{i},x_{j}\in X_{s} \end{array}\\ \begin{array}{cc} \frac{1}{n_{T}^{2}} & x_{i},x_{j}\in X_{T} \end{array}\\ \begin{array}{cc} -\frac{1}{n_{S}n_{T}} & otherwise \end{array} \end{array}\right.$$

The maximal mean difference is to be minimized in infinite-dimensional RKHS space. Combining kernel-based PCA, the problem for domain adaptation then reduces to:$$\begin{array}{c} min\\ W \end{array}\;tr\left(W^{T}KLKW\right)+\mu \cdot tr\left(W^{T}W\right)$$
(16)$$s.t.\;W^{T}KHKW=I_{m}$$
where $tr\left(W^{T}W\right)$ is a regular term that controls the model complexity; μ is a trade-off parameter; $I\in R^{m\times m}$ is the identity matrix; $H=I_{n_{S}+n_{T}}-\frac{1}{n_{S}+n_{T}}11^{T}$ is the centering matrix, where $1\in R^{n_{S\;}+n_{T}}$ is the column vector with all ones; and $I_{n_{S}+n_{T}}\;\in R^{\left(n_{S}+n_{T}\right)\times \left(n_{S}+n_{T}\right)}$ is the identity matrix. Defining *A* as a symmetric matrix, the Lagrangian of (16) is
(17)$$L=tr\left(W^{T}\left(\mu I+KLK\right)W\right)\;-\;tr\left(\left(W^{T}KHKW\;-\;I\right)A\right),$$

Setting the derivative of (17) with regard to W to zero, we have
(18)$$\left(\mu I+KLK\right)W=KHKWA$$

Take the former *m* eigenvector of $\left(\mu I+KLK\right){\;}^{-1}KHK$ as *W*. The transformed feature is then expressed as:(19)$$Z=W^{T}\varphi {\left(x\right)}^{T}\varphi \left(x\right)=W^{T}K$$

The process of marginal distribution adaption is presented in Algorithm 1.
**Algorithm 1.** Marginal Distribution Adaptation: 1: **Input: labeled source samples** Xs, Ts **and unlabeled target samples**
XT; several basic kernel functions Kpp=1m, *q, and C*2: **Output:** ZS, ZT3: **Initialize:**
γ=γ0 and t=04: **repeat**5:   Compute K⋅,⋅; γ0 by solving (1)6:   Update ||β˜p||F2 by solving (11)7:   Update γt+1 by (13)8: **until**
max{γt+1 − γt} ≤ ε9: Compute K⋅,⋅; γ with the obtained γ by solving (1)10: Compute the eigenvector of μI+KLK−1KHK11: Take the former *m* eigenvector as *W*12: Compute  ZS and ZT with (19).

### 2.2. Random Forest

Random forest [36] is a type of ensemble learning, which is to combine several base classifiers to obtain a strong classifier with significantly superior classification performance. The principle of random forest is to obtain the final classification result by voting. The generation process of random forest is shown in Figure 5.

Suppose that there is a training set T consisting of N samples (i.e., $T=\left\{t_{i}\right\},\;i=1,\;2,\;\ldots ,\;N$) and the corresponding feature vector F with M dimensions (i.e., $F=\left\{f_{j}\right\},\;j=1,\;2,\;\ldots ,\;M$). We applied a random forest with k decision trees, and the training steps are as follows:

Resample randomly from the training set based on bootstrap to form a training subset $T_{k}$;Randomly extract m features from F of $T_{k}$ without replacement ($m=\log_{2}M$ is set in this paper) to generate a complete decision tree $S_{k}$ without pruning;Repeat the above two steps k times to generate k decision trees, and then combine all of the decision trees to form a random forest;Take the test sample as the input of the random forest, and then vote on the result of each decision tree based on majority voting algorithm to obtain the classification result.

## 3. Results

We validated our method by an open-access dataset, namely the BMI dataset (http://gigadb.org/dataset/view/id/100542/ (accessed on 16 May 2021)). The research was provided by Lee et al. [8].

### 3.1. Experiment Materials and Preprocessing

The BCI system analyzed was based on Brain Amp, which utilizes 62 Ag/AgCI electrodes [8]. The EEG signals were collected at a frequency of 1000 Hz, and electrodes were placed in accordance with the international 10/20 system standard. Fifty-four subjects participated in this experiment, and none had a history of mental illness or psychoactive drugs that would affect the results of the study.

MI-BCI was tested with a dichotomous experiment in which subjects imagine their left and right hands in accordance with the directions of arrows, as shown in Figure 6. The EEG signals were recorded in two different sessions on different days. For all blocks of a session, black fixation was displayed on the screen for 3 s before each trial task began. The subjects then imagined they were performing the hand-grabbing action (grasping) in the direction specified by the visual cue. After the task, the screen display was blank for 6 s to allow the subjects to rest. There were 200 trials in one experiment per subject, half on the left and half on the right.

To retain the EEG information of the subjects as much as possible, as shown in Figure 7, 20-channel EEG data of the motor cortex region were selected: {FC5, FC3, FC1, FC2, FC4, FC6, C5, C3, C1, Cz, C2, C4, C6, CP5, CP3, CP1, CPz, CP2, CP4, CP6}. The EEG signals were downsampled to 100 Hz, and the 5th order Butterworth digital filter was utilized to obtain 8–30 Hz signals. Then, the range of 500–3500 s after the task started was selected. A common spatial pattern (CSP) was applied to maximize the difference between the two types of tasks. The first five dimensions of the feature vector were selected, and after that, the log-variance feature was calculated. Therefore, the CSP feature of a single trial was 1 × 10.

### 3.2. Model Generation

We verified the effect of the proposed domain adaption framework that combined MK-ELM and MK-MMD (denoted as MK-DA) on the aforementioned open dataset. First, the following three base kernel functions were chosen to form multiple-kernel ELM:Polynomial kernel function
(20a)$$K\left(x,y\right)={\left(x\cdot y+a\right)}^{d},d=1,2,\ldots ,N.$$

Gaussian kernel function


(20b)
$$K\left(x,y\right)=\exp(\frac{-x\;-\;y^{2}}{2\sigma^{2}}),$$


Translation-invariant of wavelet kernel function


(20c)
$$K\left(x,y\right)=\prod h(\frac{x-y}{w_{a}})\;h\left(x\right)=cos\left(w_{b}x\right)exp(-\frac{x^{2}}{w_{c}})$$


The relaxation coefficient C of the classifier was selected from $C\in \left\{0.001,\;0.01,\;0.1,\;1,\;10,\;50,\;100\right\}.$ The optimal parameters $p_{a}$ and $p_{d}$ of the poly-kernel function were selected from $p_{a}\in \left\{0.001,\;0.01,\;0.1,\;1,\;10,\;50,\;100\right\}$, and $p_{d}\in \left\{2,\;3,\;4\right\}$, respectively. The optimal parameter of the Gaussian kernel function was determined from $\sigma \in \left\{0.001,\;0.01,\;0.1,\;1,\;10,\;50,\;100\right\}$. The optimal parameters $w_{a}$, $w_{b}$, and $w_{c}$ of the wavelet kernel function were searched from w∈{0.001, 0.01, 0.1, 1, 10, 50, 100}, respectively.

Then, random forest was used as the classifier. The number of decision trees was selected from $k\in \left\{10,\;20,\;50\right\}$.

### 3.3. Experimental Results

#### 3.3.1. Methods for Comparison

This study primarily addressed the domain shift problem of BCI illiteracy by applying an open dataset containing BCI illiterate subjects [8] for validation. The classic method in which CSP was applied to extract features and linear discriminant analysis (LDA) was applied to classify, which was used as the reference framework.

The performance of the proposed DA method was compared with the performance of the DA methods that are widely used and known to achieve good results. At the same time, RF was employed as the classifier, which was the proposed method in this paper. To ensure fairness in the comparison, we gave the same parameter to all parts using the same operation, and the other parameters were given the optimal value according to the suggestions in the literature. The comparison DA methods were as follows:SA: We set the parameters referring to the research by Xiao et al. [37]. Considering the poor classification effect of BCI illiteracy, we set the subspace dimension of principal component analysis (PCA) to all to avoid information loss;GFK: We referred to the research by Wei et al. [38]. We determined the optimal dimension of the subspace by adopting the subspace disagreement measure (SDM) after the source domain and target domain data were determined;CORAL: Referring to the research by He et al. [12], we conducted a distributed computation on the feature covariance matrix of each domain and then minimized the distance between the covariance matrices of different domains;TCA: We referred to the research by Jayaram et al. [39]. In this experiment, when carrying out a multiple-kernel linear combination, the weight of the Gaussian kernel was generally the largest. Therefore, we chose the Gaussian kernel function and set its parameters to be the same as those of the Gaussian kernel function in MK-ELM;MKL: Referring to the research by Sun et al. [19] and Dai et al. [20], we combined Gaussian-kernel-based support vector machine (SVM) with MKL and applied the classifier-based DA method to optimize the target function of SVM, while minimizing the inter-domain offset based on MKL. MKL uses the three kernels above-mentioned and applied the second-order Newton method recommended by Sun et al. [18] to obtain the combination coefficients. The balance parameter was λ = 0.5. Note that the combined coefficients obtained by this method can be different from those obtained by the method proposed in this paper.

Then, the performance of the proposed classifier was compared with the performance of classifiers that are widely used in MI-BCI. The comparison classifiers were as follows:LDA: The reference method proposed by Lee et al. [8].SVM: We referred to the research by Lotte et al. [40]. We chose the Gaussian kernel function and set its parameters to be the same as those of the Gaussian kernel function in MK-ELM;KNN: We referred to the research by Lotte et al. [40]. We set the number k=5.EEGnet: We referred to the research by Lawhern et al. [41]. We set the number of channels as 20.FBCNet: We referred to the research by Mane et al. [42]. We set C as 20.

#### 3.3.2. Performance of the Domain Adaption and Classification Framework

We set the threshold value as 0.05, so *p* ≤ 0.05 indicates the statistical significance. During the experiment, according to the classification results obtained from the literature and the definition of BCI illiteracy, the subjects were divided into the following two groups:BCI (the classification result was greater than 70% in both sessions), denoted as BNI;BCI illiteracy (the classification result was less than 70% in both sessions), denoted as BI.

We performed experiments from two perspectives (i.e., cross-subject experiment and cross-session experiment). For preciseness, the Kruskal-Wallis test was adopted to display the statistical significance of the differences between methods.

**1.** 
**Results of Cross-Subject Experiments**


To ensure the simplicity of the comparison factors, both the source and target domains in this part were of the same session. Based on NBI and BI grouping, we randomly selected one subject as the source domain and another subject in the same session as the target domain in the following two ways. The first method was random sampling limited to NBI, and the second method was random sampling limited to BI. We used the proposed method DA and the control method to align the marginal distribution and employed RF as the classifier. Then, we applied different classifiers to the features adapted by MK-DA for the classification and comparison. The experiment was repeated 30 times, and the average accuracy was taken as the result. The results including the average classification accuracies (mean), standard deviation (Std), and confidence interval under 95% signification level (CI) are shown in Table 1 and Table 2.

**2.** 
**Results of the Cross-Session Experiments**


The experiments of the two sessions of each subject were taken as the source domain and the target domain, respectively. The results were divided into the BI group and the NBI group, and the average value of each group was taken as the result of that group. Then, the data of the source domain and the target domain were switched. The experimental verification was conducted in the same way as the cross-subject experiments, and the results including the average classification accuracies (mean), standard deviation (Std), and confidence interval under 95% signification level (CI) are shown in Table 3 and Table 4.

## 4. Discussion

Based on the above experimental results, the rationality of the proposed method is discussed from two perspectives: cross-subject experiment and cross-session experiment.


**Cross-Subject Experiments**


We randomly selected one subject as the subject of the training field and another subject in the same session as the target domain based on two methods, namely, the first subject was selected from those whose target domain was specified as the NBI group, and the second subject was selected from those whose target domain was specified as the BI group. When applying the proposed MK-ELM for classification, compared with LDA as the reference method, the average classification accuracies were improved, among which the biggest gain was 1.26% and 1.18%, respectively, which proved that the MK-ELM adopted in this paper increased the data divisibility in the new RKHS. Then, the competitive marginal distribution method was applied to the feature distribution adaptation. As can be seen from Figure 8, the classification accuracies of MK-DA-RF improved by 2.65% and 3.72%, respectively, compared with LDA as the reference method and by 0.78% and 0.46% compared with TCA-RF, which was the best-performing control method.


**Cross-Session Experiments**


The two sessions of the same subject were chosen as the source domain and target domain, respectively. From the results of the average classification accuracy of all subjects, the classification accuracies of the proposed MK-DA-RF improved by 6.24% and 5.74%, respectively, compared with LDA as the reference method and by 0.72% and 1.31% compared with TCA-RF, which was the best-performing control method. Then, we averaged the subjects according to the NBI and BI groups. The results of group NBI and group BI are displayed in Figure 9 and Figure 10, respectively. It can be seen that the classification accuracy of MK-ELM gained an average increase of 3.93% in the two tasks compared with the reference method for the BI group, but an average increase of 2.06% for the NBI group. Since the subjects in the BI group could not effectively control the BCI, the extracted features were difficult to distinguish. The divisibility was increased after features were mapped to multiple-kernel-based RHKS by MK-ELM, which was consistent with the experimental phenomenon from the results. Then, under the adjustment of DA, the combined method of MK-ELM and MK-MMD proposed in this study significantly improved the classification accuracy. In particular, in the BI group, the average classification accuracies of MK-DA-RF were 5.9% and 6.3% higher than those of the reference method and 0.62% and 1.34% higher than those of TCA-RF, which was the best-performing control method.

The feature distribution of the same subject before and after MDA was observed, and the first two dimensions of the feature value were taken as the *X*-axis and *Y*-axis, respectively, to check the feature distribution. Figure 11 shows the original feature distribution of Subject 19 and the feature distribution obtained with the method proposed in this study. Specifically, Figure 11a,b is the feature space distribution of class 1 (left-hand motor imagery) and class 2 (right-hand motor imagery), respectively. It can be seen that the class divisibility of the feature distribution was increased with MK-DA. Figure 11c,d refers to the feature distribution before and after MK-DA was employed. It can be seen that the feature space of the source domain and the target domain were further closer by the method proposed in this paper.


**Performance of Random Forest**


In this section, we analyzed the performance of different classifiers separately in two experiments. The performance evaluation metrics were calculated referring to the research by Giannakakis et al. [43].

In the cross-subject experiment, as shown in Figure 12, RF achieved relatively better results in all experiments. The classification accuracies of the proposed RF improved by 0.17% and 0.17%, respectively, compared with LDA, which was the best-performing control method. The results of the performance evaluation metrics are shown in Table 5. The confusion matrices and receiver operating characteristic curves are shown in Figure 13 and Figure 14.

In the cross-session experiment, the results of all subjects are displayed in Figure 15. It can be seen that the classification accuracy of RF gained an average increase of 0.24% and 0.58%, respectively, in the two tasks compared with the control method with the best performance. The results of the performance evaluation metrics are shown in Table 6. The confusion matrices and receiver operating characteristic curve are shown in Figure 16 and Figure 17.

In particular, in all experiments, the performance of EEGnet was worse than that of the LDA without domain adaptation, so we believe that it is related to the fact that 20 channels were used, but the training data were so small that the model overfitted.


**Computational Complexity**


Let $l_{S}$ and $l_{T}$ denote the number of training samples and testing samples, respectively, and each sample $x_{i}\in R^{d}$. Suppose we grow k trees for RF. The computational complexity of each step is shown in Table 7, which is based on Liu et al. [26], Pan et al. [27], and Biau [36], where $t_{\gamma }$ is the maximum number of iterations and m is the number of base kernels. Therefore, the computational complexity of MK-DA is $t_{\gamma }*O\left(1\right)+m*l_{S}^{2}*O\left(d\right)+O(d\cdot {\left(l_{S}+l_{T}\right)}^{2})$. Then, the computational complexity of the proposed framework is $t_{\gamma }*O\left(1\right)+m*l_{S}^{2}*O\left(d\right)+O(d\cdot {\left(l_{S}+l_{T}\right)}^{2})+O\left(k\cdot \left(d\cdot l_{S}\cdot \log l_{S}\right)\right)$.


**Limitations**


However, there are also problems with the proposed method. First, after the features were adapted, the classification accuracies applying RF in both types of experiments were higher than the LDA after domain adaption, that is, the average classification accuracy of RF was 0.17% higher than that of LDA in the cross-subject experiment, and was 0.41% higher than that of the LDA in the cross-session experiment. However, the computational complexity of RF was significantly higher than that of the LDA. Therefore, it is necessary to combine classification accuracy with the computational complexity in selecting the suitable classifier. Second, parameters involved in the proposed framework (i.e., the relaxation coefficient C of ELM, the initial parameters of the kernels) were all selected from among a limited number of values, but the choice of the parameters would affect the classification results. Therefore, optimization methods will be suggested to solve this problem in our subsequent studies.

## 5. Conclusions

The method proposed in this paper aimed to address the inter-domain differences of EEG-based motor imagery BCI, especially for BCI illiteracy. It was found that BCI illiterate subjects could not effectively control the BCI due to two major problems: difficulties in classifying and poor repeatability. We proposed a domain adaption method that combines MK-ELM and MK-MMD. To demonstrate the effectiveness of the method, we performed experiments from two perspectives (i.e., cross-subject experiment and cross-session experiment). The MK-ELM achieved relatively better results than the LDA in all experiments. Meanwhile, it can be seen from the results of MK-DA that the MK-DA with each classifier achieved relatively better results in all combination forms. The average accuracies of all experiments of MK-DA combined with LDA was 3% higher than that of LDA in the cross-subject experiments, and was 5.6% higher in the cross-session experiments. Therefore, the divisibility was increased after the features were mapped to multiple-kernel-based RHKS by MK-ELM, and the domain shift decreased by MK-MMD, which was consistent with the experimental phenomenon from the results. At the same time, RF that could effectively handle high-dimensional features was employed as a classifier. It can be seen from the results of MK-DA-RF in the cross-subject experiments that the average classification accuracy of all the experiments could reach 70.4%, which was 2.7% higher than that of the reference method “CSP + LDA” and 0.3% higher than that of the best-performing control method. In the cross-session experiments, the average classification accuracy of the proposed method for all experiments could reach 73.6%, 6.1% higher than that of the reference method, and 0.4% more than that of the best-performing control method. Particularly for the BCI illiterate subjects, the average classification accuracy of all the experiments with target subjects showed that BCI illiteracy could reach 63.4% with the proposed method, which was 5.3% higher than the reference method without domain adaption. Therefore, the method proposed in this paper could achieve a significant effect in the BCI illiteracy group.

## Figures and Tables

**Figure 1 sensors-22-06572-f001:**
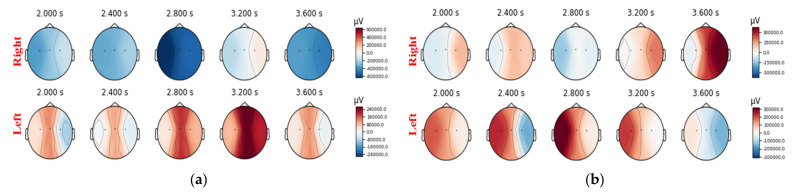
The power spectral density (PSD) of Subject 46 performing motor imagery at different times: the charts are the power spectra (blue represents negative, and red represents positive). The EEG signals were provided by an open dataset with BCI illiterate subjects [8] and they were recorded in two different sessions on different days. (**a**) and (**b**) are the PSD diagrams of Subject 46 in session 1 and session 2, respectively. The classification accuracies of Subject 46 were 53% and 58% in the two sessions, respectively, so the subject was classified as BCI illiterate.

**Figure 2 sensors-22-06572-f002:**
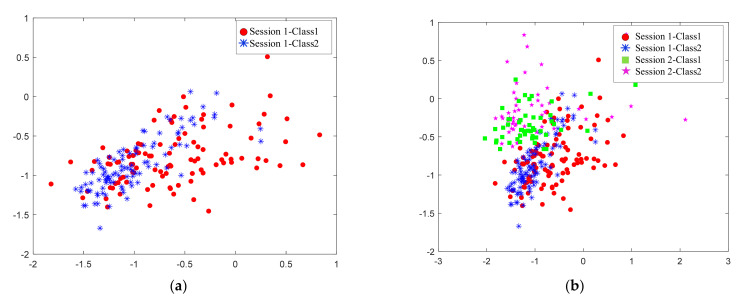
The feature distribution comparison of Subject 46. The features were extracted by a common spatial pattern (CSP). (**a**) The distribution of features for session 1; (**b**) The distribution comparison of features for Sessions 1 and 2.

**Figure 3 sensors-22-06572-f003:**
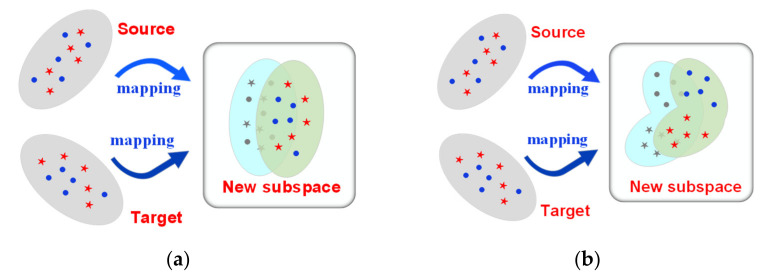
The distribution adaptation. (**a**) Purely for marginal distribution adaptation; (**b**) With the features being mapped into the new space, the inter-class maximal divisibility was achieved and the distribution difference between the source domain and target domain was reduced.

**Figure 4 sensors-22-06572-f004:**
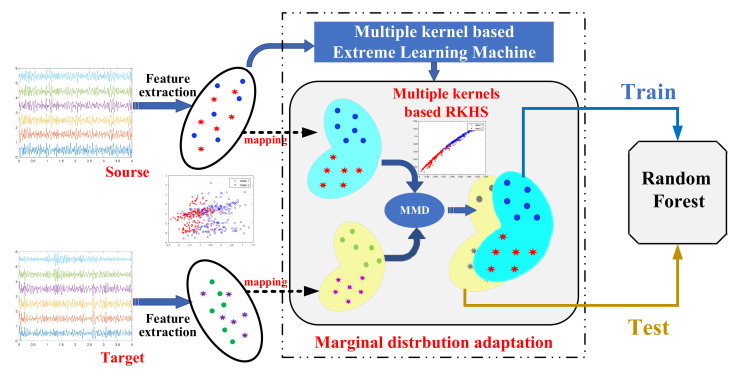
The joint distribution adaptation framework.

**Figure 5 sensors-22-06572-f005:**
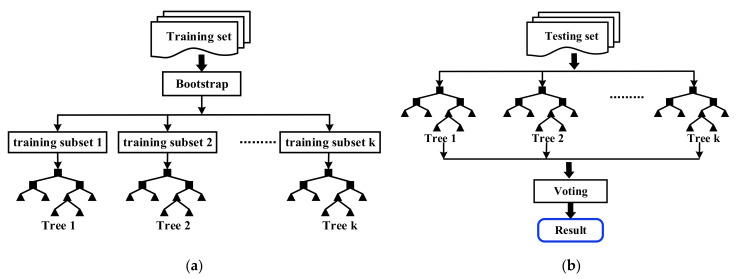
The generation process of random forest. (**a**) The generation of the forest. (**b**) The implementation of decisions.

**Figure 6 sensors-22-06572-f006:**
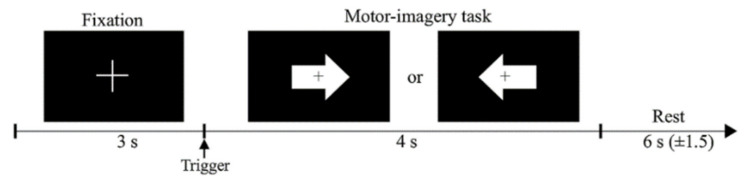
The experimental design for binary class MI. The experimental description was provided by Lee et al. [8].

**Figure 7 sensors-22-06572-f007:**
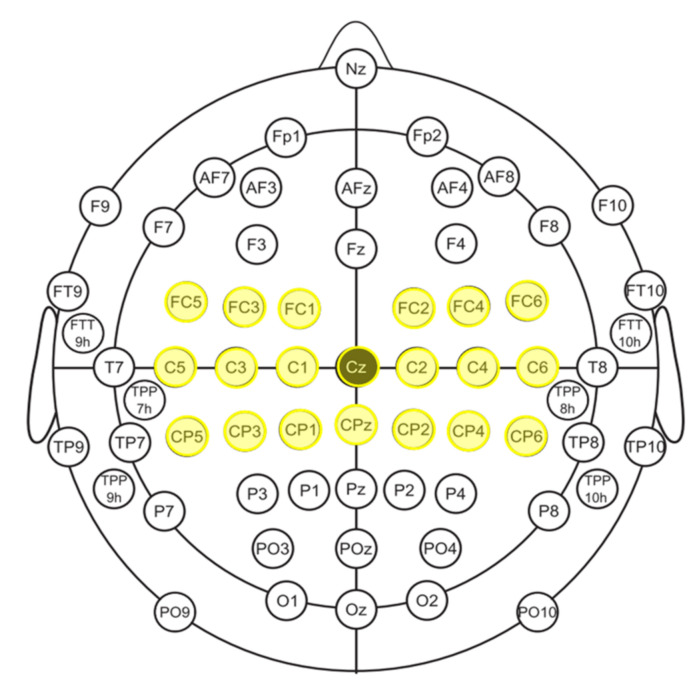
The electrode position.

**Figure 8 sensors-22-06572-f008:**
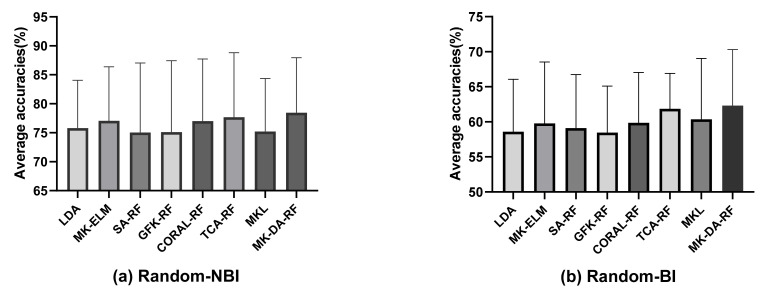
The results of the different distribution adaptation methods in the cross-subject experiments.

**Figure 9 sensors-22-06572-f009:**
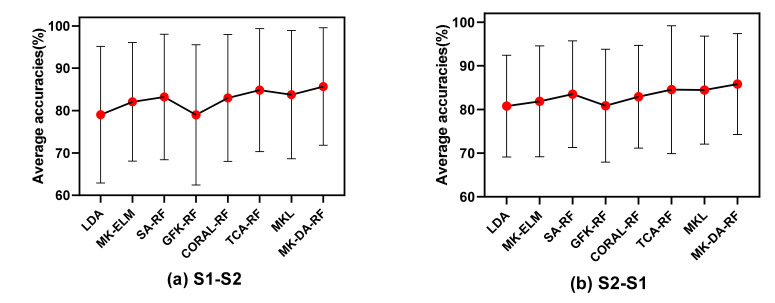
The results of the different distribution adaptation methods for the NBI group.

**Figure 10 sensors-22-06572-f010:**
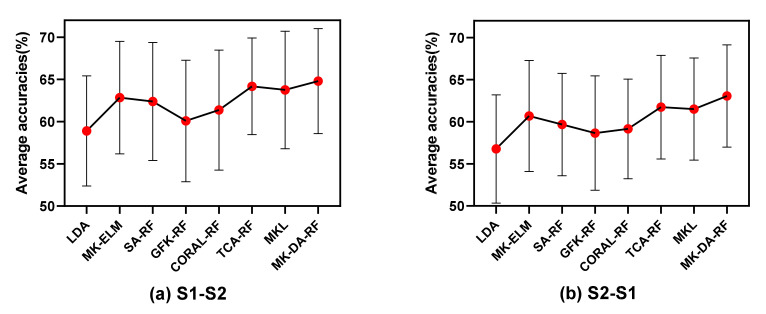
The results of the different distribution adaptation methods for the BI group.

**Figure 11 sensors-22-06572-f011:**
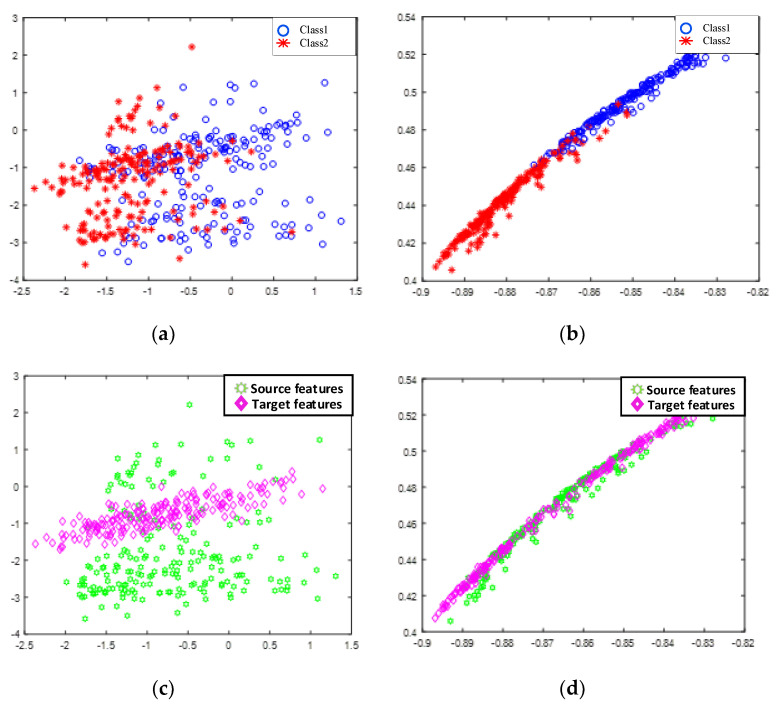
The feature space distribution of Subject 22. With the reference method, Subject 22 had a classification accuracy of 68% from session 1 to session 2. With the improved method, the classification accuracy was 83%. (**a**) The original feature space distribution of class 1 (left-hand imagery) and class 2 (right-hand imagery) in session 1; (**b**) The feature space distribution of class 1 and class 2 obtained after marginal distribution adaptation session 1; (**c**) The original feature space distribution in session 1 (source domain) and session 2 (target domain) for all categories; (**d**) The feature space distribution obtained after marginal distribution adaptation in session 1 (source domain) and session 2 (target domain) for all categories.

**Figure 12 sensors-22-06572-f012:**
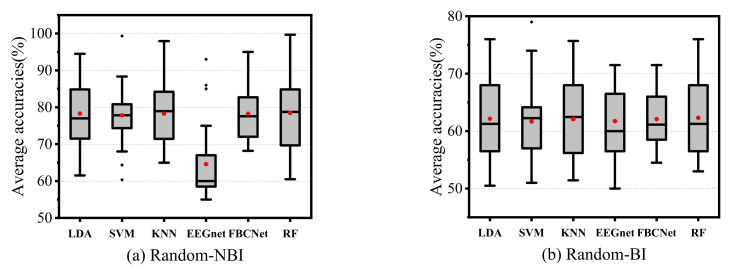
The results of different classifiers in the cross-subject experiment.

**Figure 13 sensors-22-06572-f013:**
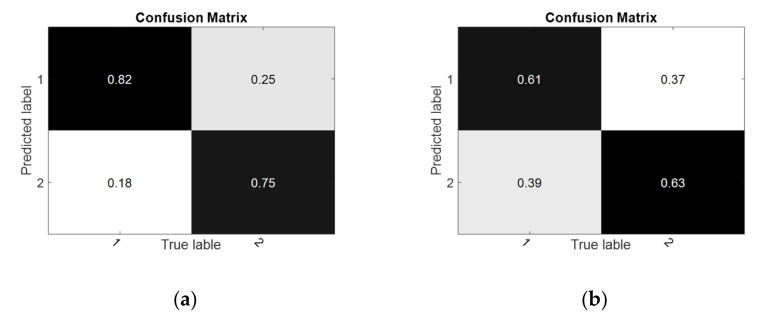
The confusion matrices of RF in the cross-subject experiment. (**a**) Random-NBI; (**b**) Random-BI.

**Figure 14 sensors-22-06572-f014:**
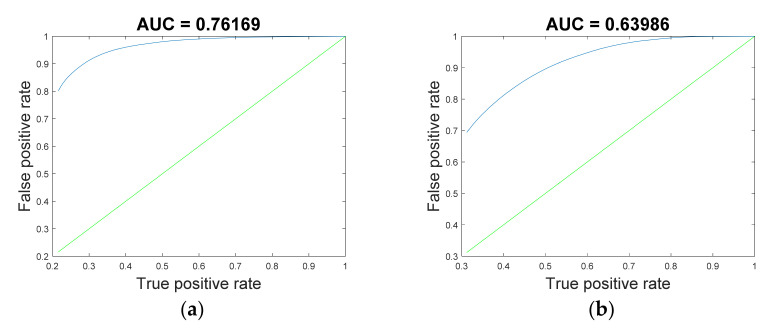
Receiver operating characteristic curve of RF in the cross-subject experiment. (**a**) Random-NBI; (**b**) Random-BI.

**Figure 15 sensors-22-06572-f015:**
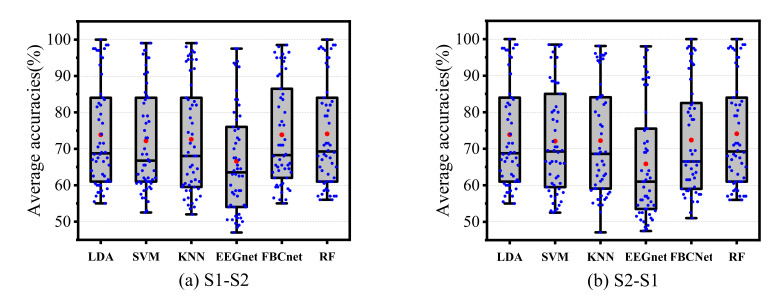
The results of the different classifiers in the cross-session experiment.

**Figure 16 sensors-22-06572-f016:**
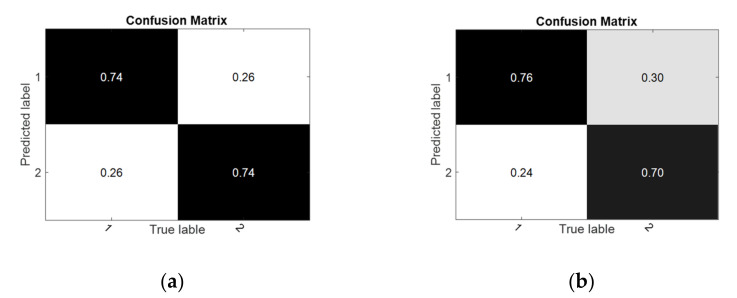
The confusion matrices of RF in the cross-session experiment. (**a**) S1-S2; (**b**) S2-S1.

**Figure 17 sensors-22-06572-f017:**
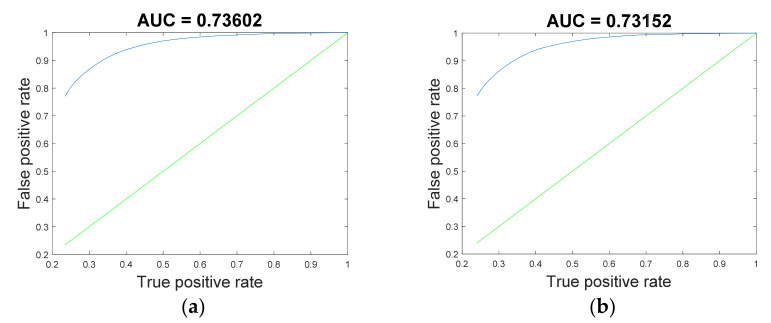
The receiver operating characteristic curve of RF in the cross-session experiment. (**a**) S1-S2; (**b**) S2-S1.

**Table 1 sensors-22-06572-t001:** A comparison of the different distribution adaptation methods for the source and target data from different subjects.

Tasks	Results	LDA **	MK-ELM *	SA-RF *	GFK-RF **	CORAL-RF *	TCA-RF *	MKL *	MK-DA-RF
**Random-NBI**	Mean	75.81	77.07	75.05	75.12	77.0	77.68	75.22	**78.46**
	Std	8.25	9.32	11.98	12.31	10.73	11.14	9.14	9.5
	CI	2.95	3.33	4.28	4.40	3.84	3.98	3.27	3.40
**Random-BI**	Mean	58.60	59.78	59.11	58.47	59.86	61.86	60.36	**62.32**
	Std	7.47	8.74	7.64	6.64	7.17	5.04	8.67	8.01
	CI	2.67	3.13	2.73	2.37	2.56	1.08	3.10	2.87

Note: * indicates *p* < 0.05, ** indicates *p* < 0.01, and there is no * when *p* > 0.05. The best results are indicated in bold text; SA-RF indicates the DA method is SA and the classifier is RF, Other interpretations are the same.

**Table 2 sensors-22-06572-t002:** A comparison of the different classifiers in the cross-subject experiments.

Tasks	Results	LDA *	SVM **	KNN **	EEGnet **	FBCNet **	RF
**Random-NBI**	Mean	78.29	77.81	78.26	64.60	78.23	**78.46**
	Std	8.48	7.40	8.63	9.77	7.60	9.50
	CI	3.03	2.65	3.09	3.49	2.72	3.40
**Random-BI**	Mean	62.15	61.67	62.13	61.72	62.09	**62.32**
	Std	7.49	6.62	6.91	8.36	4.80	6.95
	CI	2.68	2.37	2.47	2.99	1.72	2.49

Note: * indicates *p* < 0.05, ** indicates *p* < 0.01, and there is no * when *p* > 0.05. The best results are indicated with bold text.

**Table 3 sensors-22-06572-t003:** A comparison of the different distribution adaptation methods for the source and target data from different sessions.

Group	Task	Results	LDA **	MK-ELM **	SA-RF *	GFK-RF **	CORAL-RF *	TCA-RF *	MKL *	MK-DA-RF
**NBI**	**S1-S2**	Mean	79.02	82.06	83.21	78.98	82.98	84.83	83.75	**85.69**
	**S2-S1**	Mean	80.77	81.85	80.85	80.85	82.83	84.54	84.44	**85.81**
**BI**	**S1-S2**	Mean	58.90	62.85	60.08	60.08	61.37	64.18	63.75	**64.80**
	**S2-S1**	Mean	56.77	60.68	58.65	58.65	59.15	61.73	61.50	**63.07**
**ALL**	**S1-S2**	Mean	67.84	71.39	68.48	68.48	70.97	73.36	72.64	**74.08**
		Std	15.04	14.37	15.09	14.90	14.87	15.64	14.80	14.43
		CI	4.12	3.79	4.06	4.11	4.15	3.93	4.02	3.90
	**S2-S1**	Mean	67.44	70.09	68.52	68.52	69.68	71.87	71.69	**73.18**
		Std	15.43	14.22	15.21	15.40	15.57	14.72	15.05	14.64
		CI	4.01	3.83	4.03	3.97	3.97	4.17	3.95	3.85

Note: * indicates *p* < 0.05, ** indicates *p* < 0.01, and there is no * when *p* > 0.05. The best results are indicated with bold text; SA-RF indicates the DA method is SA and the classifier is RF, Other interpretations are the same.

**Table 4 sensors-22-06572-t004:** A comparison of the different classifiers in the cross-session experiments.

Tasks	Results	LDA *	SVM *	KNN **	EEGnet **	FBCNet **	RF
**S1-S2**	Mean	73.84	72.18	72.61	66.56	73.84	**74.08**
	Std	14.50	14.19	15.13	14.49	14.10	14.43
	CI	3.87	3.78	4.04	3.86	3.76	3.85
**S2-S1**	Mean	72.60	72.12	72.22	65.84	72.36	**73.18**
	Std	14.23	14.62	15.20	15.43	14.91	14.64
	CI	3.87	3.90	4.06	4.12	**3.98**	**3.91**

Note: * indicates *p* < 0.05, ** indicates *p* < 0.01, and there is no * when *p* > 0.05. The best results are indicated with bold text.

**Table 5 sensors-22-06572-t005:** The performance matrices of RF in the cross-subject experiments.

Task	Kappa	Recall	F1-Score	Precision	AUC
**S1-S2**	0.703	0.816	0.791	0.768	0.762
**S2-S1**	0.451	0.613	0.626	0.619	0.640

**Table 6 sensors-22-06572-t006:** The performance matrices of RF in the cross-session experiments.

Task	Kappa	Recall	F1-Score	Precision	AUC
**S1-S2**	0.631	0.743	0.741	0.739	0.736
**S2-S1**	0.623	0.760	0.738	0.717	0.732

**Table 7 sensors-22-06572-t007:** The computational complexity of each step.

Step	Computational Complexity
MK-ELM	$t_{\gamma }*O\left(1\right)+m*l_{S}^{2}*O\left(d\right)$
TCA	$O(d\cdot {\left(l_{S}+l_{T}\right)}^{2})$
RF	$O\left(k\cdot \left(d\cdot l_{S}\cdot \log l_{S}\right)\right)$

## Data Availability

We validated our method by an open-access dataset, namely, the BMI dataset (http://gigadb.org/dataset/view/id/100542/ (accessed on 16 May 2021)). Thus, our study did not involve any self-recorded datasets from human participants or animals. The authors declare that they have no conflicts of interest.

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
