# Peer review of "Distribution Adaptation and Classification Framework Based on Multiple Kernel Learning for Motor Imagery BCI Illiteracy"

_sensors, 2022, doi:10.3390/s22176572_

Round 1

Reviewer 1 Report

 Thank you for your manuscript. I like this work and I found it very interesting as expert in the field.

The authors addressed MI-BCI illiteracy problem by proposing a distribution adaptation method. It is based on multiple-kernel learning to make the distribution of features between the source and target domain closer to each other and maximize categories' divisibility. A multiple-kernel-based extreme learning machine is adapted to train the labeled data in the source domain to find a new subspace and eliminate the difference in feature distribution between domains in the new subspace. The paper is very well written and organized.

However, I have some comments which will be fixed to improve the quality of the paper.

  1. The authors mentioned in line 35, "As shown in Figures 1 and 2, Power Spectral Density (PSD) of subject 46, who is BCI illiterate, are quite different in each of the two sessions." This is always true from a neuroscience viewpoint and is not new [2]. It is well known as that the maps is differed for the same subject and during the movement of the same limbs, and from one hour to another [3]. 
  2. It would be very appreciated to add the preprocessing effect to the your approach. As shown in [1], the classification accuracy is highly sensitive to the filter block. The idea is to show your method's efficiency against the filter block's tuning. For example, you can use an FIR filter with different order before the feature extraction and see the variation of the results.  
  3. In line 87, the author mention, "The combined kernel function could still satisfy the Mercer condition." Can you write in two or three sentences the Mercer condition? 
  4. The transition from equations 5 to 6 is unclear. 
  5. In figure 6, please add a citation to the figure caption. 
  6. The authors mentioned the use of 20 channels from 64 channels. The selection of electrodes is based on what? Did the authors study the effect of the channel's selection on the accuracy. 
  7. For the computational complexity (Line 450), please reorganize it by using equations or put them in a table to be more apparent for the reader.
  8. As you are using a public dataset, can you add a comparison table with the last published studies that use the same dataset? 
  9. I suggest adding the std of the classification accuracy in Tables 2 and 3. 

[1] 10.1109/CCMB.2014.7020704

[2]10.1371/journal.pbio.2000106

Reviewer 2 Report

Two points are request strongly before further processing:

1. Confidence interval(CI) under 95% signification level

2. The following performance matrices need for justification of the scheme

a.  Fleiss Kappa or Cohen’s Kappa

b. Recall/Sensitivity/Sensibility

c. F1-score or measure.

d.  ROC curve (Receiver operating characteristic curve)

e.  AUC (Area under the ROC Curve)
